# Bacterial Magnetosomes Release Iron Ions and Induce Regulation of Iron Homeostasis in Endothelial Cells

**DOI:** 10.3390/nano12223995

**Published:** 2022-11-13

**Authors:** Wenjia Lai, Dan Li, Qingsong Wang, Yan Ma, Jiesheng Tian, Qiaojun Fang

**Affiliations:** 1Division of Nanotechnology Development, National Center for Nanoscience and Technology, Chinese Academy of Sciences, Beijing 100190, China; 2Laboratory of Theoretical and Computational Nanoscience, National Center for Nanoscience and Technology, Chinese Academy of Sciences, Beijing 100190, China; 3State Key Laboratory of Protein and Plant Gene Research, College of Life Sciences, Peking University, Beijing 100871, China; 4Aviation Service Department, Yantai Engineering & Technology College, Yantai 264006, China; 5State Key Laboratories for Agrobiotechnology, College of Biological Sciences, China Agricultural University, Beijing 100193, China

**Keywords:** magnetosome, endothelial cell, degradation, quantitative proteomic, iron metabolism

## Abstract

Magnetosomes (MAGs) extracted from magnetotactic bacteria are well-defined membrane-enveloped single-domain magnetic nanoparticles. Due to their superior magnetic and structural properties, MAGs constitute potential materials that can be manipulated via genetic and chemical engineering for use in biomedical and biotechnological applications. However, the long-term effects exerted by MAGs on cells are of concern in the context of in vivo applications. Meanwhile, it remains relatively unclear which mechanisms are employed by cells to process and degrade MAGs. Hence, a better understanding of MAGs’ degradation and fundamental signal modulations occurring throughout this process is essential. In the current study, we investigated the potential actions of MAGs on endothelial cells over a 10-day period. MAGs were retained in cells and found to gradually gather in the lysosome-like vesicles. Meanwhile, iron-ion release was observed. Proteomics further revealed a potential cellular mechanism underlying MAGs degradation, in which a group of proteins associated with vesicle biogenesis, and lysosomal enzymes, which participate in protein hydrolysis and lipid degradation, were rapidly upregulated. Moreover, the released iron triggered the regulation of the iron metabolic profiles. However, given that the levels of cell oxidative damage were relatively stable, the released iron ions were handled by iron metabolic profiles and incorporated into normal metabolic routes. These results provide insights into the cell response to MAGs degradation that may improve their in vivo applications.

## 1. Introduction

Magnetic iron oxide nanoparticles are one of the most popular nanoparticle formulations used in a wide range of biomedical and biotechnology applications [1,2]. They have proven particularly useful as magnetic resonance imaging (MRI) contrast agents and magnetic hyperthermia agents [3,4]. Bacterial magnetosomes (MAGs), a type of biologically synthesized magnetic nanoparticles found in most magnetotactic bacteria, consist of magnetite (Fe_3_O_4_) or greigite (Fe_3_S_4_) nanoparticle cores and are naturally covered by a protein-rich phospholipid bilayer, which is homologous to bacterial cell membranes [5]. Due to the complex genetically controlled biomineralization process, mature MAGs are colloidally stable particles, with high magnetism and narrow size distribution. MAG crystals are generally 35–120 nm in diameter, indicating that they are ferrimagnetic with a single domain [6,7]. These unique features have attracted tremendous interest in biomedical and biotechnological research. Indeed, isolated MAGs have been successfully tested in hyperthermia and imaging techniques, such as MRI and magnetic particle imaging [8,9]. In addition, the protein-rich phospholipid bilayer of MAG provides a variety of functional groups that allow for controllable functionalization of the particle surface via chemical and genetic manipulation. Such modified MAGs are currently under investigation for specialized sensors and drug delivery [5,10].

The broad potential application of MAGs in a myriad of fields has prompted the scientific community to investigate the associated biocompatibility, as well as the mechanisms underlying their potential cellular effects. Although MAG toxicity has been assessed at different concentrations and exposure times, using different mammalian cell types (in vitro) and in vivo models [11,12,13,14,15,16], the biotransformation of MAG in cells is still not well elucidated. Unlike artificial iron oxide nanoparticles with chemically modified surfaces, MAGs encapsulate the magnetic crystal with a membrane and proteins that provide a natural “coating”. As a naturally synthesized nanoparticle with a natural interface when interacting with mammalian cells, little is known about MAG degradation and subsequent biological responses occurring within the mammalian cell. Understanding how mammalian cells respond to MAG and whether these cells degrade and/or assimilate MAG is important to ensure MAG further in vivo applications.

In vitro and in vivo studies have shown that nanoparticle uptake can be achieved by many mammalian cell types, including endothelial cells (ECs), B cells, macrophages, and Kupffer cells [17,18,19]. ECs constitute the luminal lining of all vascular system components and form the endocardium. The endothelial monolayer lining blood vessels tightly regulates the exchange of nutrients between the blood and surrounding tissues [20] and serves as one of the main cell types that interacts with nanoparticles [21]. In particular, in vivo investigations of MAGs in cancer treatment require intravenous injection [5], through which they are readily exposed to the bloodstream and can interact with proteins, blood cells, and ECs. Recently, our group established that MAGs from *Magnetospirillum gryphiswaldense* MSR-1 form protein corona after interaction with the human plasma and uptake by ECs [22]. Meanwhile, in the current study, we applied quantitative proteomics to assess the effect of MAGs on ECs over a 10-day period by observing their uptake, distribution, degradation, and subsequent cell responses at the molecular level. We show here that the MAGs are transported to lysosome-like vesicles and retained by cells throughout the time course. The MAGs induce ECs to upregulate vesicle-biogenesis-related proteins and the expression of lysosomal enzymes. Furthermore, the released iron ions of MAGs subsequently trigger the regulation of ECs’ iron homeostasis, which, in turn, avoids free iron accumulation and excess oxidative damage in ECs. Overall, this study suggests that, at low dose of MAGs treatment, ECs slowly degrade MAGs and maintain homeostasis, indicating that MAGs are biocompatible, as well as useful for living organisms to biotransform them after administration. We believe that the findings will help to facilitate the future use of bacterial MAGs in biomedical and biotechnological applications.

## 2. Materials and Methods

### 2.1. Bacterial Strain and Magnetosome Preparation

*Magnetospirillum gryphiswaldense* MSR-1 (DSM6361) was a kind gift from China Agricultural University and was cultured at 30 °C, 110 rpm [23]. MSR-1 cells and MAGs were harvested by using PBS with 0.5% Tween 20, as described previously [22]. After purification, MAGs were washed twice and suspended in ultrapure water, lyophilized, disinfected by using UV light, and stored at −80 °C. For further assays, MAGs were diluted in ultrapure water, and the Fe concentration was determined by using o-phenanthroline spectrophotometry [22].

### 2.2. Examination of Purified MAGs and Amine Iron Oxide Nanoparticles

Amine magnetic iron oxide nanoparticles (Product ID: SHA-30, 5 mg/mL Fe) (IONPs) were purchased from Ocean NanoTech (San Diego, CA, USA). The purified MAGs and IONPs were suspended in 20 μL of ultrapure water and placed onto the carbon-coated copper grids. Negative staining was performed to assess the membranes of purified MAGs, as previously described [22]. Samples were observed by using transmission electron microscopy (TEM; Tecnai G2 F30 S-TWIN, Thermo Scientific, Waltham, MA, USA). To assess the protein composition of isolated MAGs, membrane proteins were extracted, and the in-solution two-step digestion procedure was performed, as described [24].

Corona formation on MAGs and IONPs were established as previously described [22]. The protein content of MAGs with or without corona and IONPs with corona were measured by using BCA Protein Assay Kit (Beyotime, Jiangsu, China). Zeta potentials of MAGs and IONPs with or without corona were measured in water or post-exposure to human plasma (Product GTX73265, GeneTex, Taiwan, China) by Malvern Zetasizer Nano ZS (Malvern Instruments, Worcestershire, UK), at 25 °C, as previously described [22].

### 2.3. Cell Culture and Treatment

Human normal vascular ECs were bought from CHI Scientific Inc.(CHI-1-0028, Maynard, MA, USA) and grown in Dulbecco’s Modified Eagle Medium: Nutrient Mixture F-12 (DMEM/F-12) supplemented with 10% fetal bovine serum (Gibco, Grand Island, NY, USA), 10% endothelial cell growth supplement (ScienCell, Carlsbad, CA, USA), 20 ng/mL recombinant human vascular endothelial growth factor (Novoprotein, Jiangsu, China), 50 IU/mL penicillin, and 50 mg/mL streptomycin (Invitrogen, Carlsbad, CA, USA), at 37 °C, in an atmosphere of 95% humidified air and 5% CO_2_. The medium was changed every 3 days.

After ECs were grown into a confluent monolayer, they were washed twice with phosphate buffer saline (PBS, Macgene, Beijing, China). MAGs or IONPs with corona were suspended in conditioned medium containing DMEM/F-12 without phenolsulfonphthalein (Gibco BRL, Grand Island, NY, USA), 50 IU/mL penicillin, and 50 mg/mL streptomycin and added to the cell. The same volume of H_2_O in the conditioned medium was set as the control. Cells were further cultured at 37 °C. After 48 h, the medium was removed, and cells were washed with PBS thrice; then fresh conditioned medium without nanoparticles was added and changed every 3 days.

### 2.4. Cell Toxicity Assay

ECs grown in 96-well plates were treated with or without nanoparticles at the designated times. Cells toxicity was analyzed via crystal violet staining. The culture medium was aspirated, and the cells were gently washed with PBS and fixed by using 4% polyoxymethylene for 15 min and further stained with 0.1% crystal violet solution (Beyotime, Jiangsu, China) for 20 min. Cell stain was washed off five times with PBS, after which the crystal violet stain was dissolved in 33% acetic acid and the absorbance was detected at 590 nm. The background of 96-well plates with cell and nanoparticles was also measured by adding fresh 33% acetic acid solution after discarding the dissolved supernatant and washing the plate five times with PBS. The cell viability was calculated by subtracting the background and normalizing the signal to the control (cell treated without nanoparticles).

### 2.5. Flow Cytometry Analysis of the Cell Cycle

ECs cultured in 12-well plates were treated with or without nanoparticles for 48 h. Cells were then harvested via trypsin digestion, washed twice with cold PBS, and fixed with cold 70% ethanol at -20 °C, overnight. After centrifugation at 1000× *g* for 5 min, the fixed cells were subsequently resuspended in 0.2% Triton X-100 and 10 mg/mL RNaseA in 0.5 mL PBS at 37 °C for 30 min. Then the cells were stained with 5 μg/mL propidium iodide at 4 °C for 20 min and suspended in 1 mL 0.1% BSA. Finally, the cells were analyzed via flow cytometry, and the DNA content was quantified by using ModFit LT software (Verity Software House: Augusta, Topsham, ME, USA).

### 2.6. Determination of Lipid Peroxidation and Protein Oxidation

Lipid peroxidation marker malondialdehyde (MDA) content was determined by using the Lipid Peroxidation MDA Assay Kit (Beyotime, Jiangsu, China), according to the manufacturer’s instructions. The treated ECs were extracted by using the lysis buffer (6 M guanidine hydrochloride, 50 mM Tris, pH 8.0), followed by centrifugation. The supernatant was collected and mixed with the MDA detection solution, boiled, and centrifuged, and then 200 μL of supernatant was transferred to a 96-well plate. The absorbance was measured at 532 nm, and the MDA content was calculated according to the standard curve.

The protein oxidation level was estimated by measuring the protein carbonyl content [25], using the Protein Carbonyl Colorimetric Assay Kit (Cayman Chemical, Ann Arbor, MI, USA). Proteins from treated ECs were extracted, using the lysis buffer. After centrifuging at 10,000× *g* for 15 min at 4 °C, the supernatant was collected, and the assay was performed according to the manufacturer’s instructions. The absorbance was measured at 370 nm.

The protein concentration of each sample was determined by using a 2D Quant kit (GE Healthcare, Chicago, IL, USA). The MDA and protein oxidation levels were normalized to the protein amount of each sample.

### 2.7. Intracellular Localization of MAGs and IONPs and Iron Quantification by Inductively Coupled PlasmaMass Spectrometry (ICP-MS)

The cultured cells were treated with nanoparticles for 48 h (10 μg/mL Fe concentration), washed twice with PBS, fixed using 2.5% glutaraldehyde solution for 2 h, and then postfixed for 1 h in 1% osmium tetroxide in deionized water. After dehydration in increasing concentrations of ethanol (from 70% up to 100%), the samples were immersed in an ethanol/Epon (1:1 *v*/*v*) mixture for 1 h before being transferred to pure Epon and embedded at 37 °C for 2 h. The final polymerization was carried out at 60 °C for 24 h. Ultrathin sections (50 nm) were mounted on copper grids and stained with uranyl acetate and lead citrate before being examined by using TEM (FEI Tecnai G2 20 Twin, Thermo Scientific, Waltham, MA, USA).

Prussian Blue Staining Kit (Leagene Biotechnology, Beijing, China) was applied to detect intracellular iron content according to the manufacturer’s instructions. ECs grown in 24-well plates treated with MAGs or IONPs (10 μg/mL Fe concentration) were washed with PBS thrice to remove the free particles and then fixed with 4% paraformaldehyde for 15 min. The particles within the cells were stained with Prussian blue staining solution, and the cell was further stained with eosin solution. Photomicrographs of stained samples were acquired by using an Olympus IX-71 Research Inverted System Microscope (Olympus, Tokyo, Japan).

The amount of soluble iron and insoluble iron particles in cells was measured by using ICP-MS. ECs grown in 12-well plates treated with nanoparticles (10 μg/mL Fe concentration) were washed with PBS thrice to remove the free particles. Then cells were collected by trypsinization and centrifuged at 1000× *g* for 5 min. The cell pellet was washed with PBS another three times and suspended in lysis buffer. The cell lysate was further centrifuged at 20,000× *g* for 60 min; the supernatant was collected for the intracellular iron ion content analysis and the pellet for insoluble particles iron content analysis. All samples were sent for ICP-MS analysis (iCAP Qc, Thermo Scientific, Waltham, MA, USA).

### 2.8. Proteomic Analysis

Cells treated with or without MAGs were collected, lysed, digested, and labeled with Tandem Mass Tag (TMT) 10 plex™ Isobaric Label Reagent Set labeling kit (Thermo Scientific, Waltham, MA, USA) according to the manufacturer’s instructions (Appendix A). About 3 µg of mixed labeled peptide was loaded onto the Q-Exactive instrument (Thermo Scientific, Odense, Denmark) with a 240 min or 320 min gradients. To ensure data quality, for each sample three biological repeats and two technical repeats were performed. The mass spectrometry (MS) raw data from TMT labeled peptides were identified, quantified using default processing and consensus workflow for tandem MS TMT quantification method with Thermo software Proteome Discoverer (version 2.2). The peptides samples from purified MAGs were identified by using Q-Exactive instrument with a 145 min gradient. The MS raw data of peptides isolated from MAGs were submitted to the MaxQuant software (version 1.6) with label-free quantification workflow, using the Andromeda search engine. The detailed procedures and parameters are described in the Appendix A.

### 2.9. Bioinformatics Analysis

Proteomic data analysis was performed by using SPSS software (version 16.0) and R statistical computing environment [26]. Gene Ontology (GO) and pathway enrichment of differentially expressed proteins were performed by using Cytoscape plugged ClueGO + CluePedia [27,28].

### 2.10. Immunofluorescence and Western Blot Analysis

For immunofluorescence analysis, the treated cells were washed five times with PBS and fixed with 4% polyoxymethylene for 20 min, at room temperature; washed three times with PBS; permeabilized by using 1% Triton X-100 for 20 min, at room temperature; washed three times with PBST (0.05% Tween 20 in 0.1 M PBS); blocked with 10% bovine serum albumin (BSA) in PBST for 1 h at 37 °C; and then incubated with rabbit anti-human ferritin (Product ab75973, Abcam, Cambridge, UK) (diluted in a ratio of 1:100) in 1% BSA, overnight, at 4 °C. After being washed thrice with PBST, fluorescence isothiocyanide (FITC)-conjugated goat anti-rabbit immunoglobulin (IgG) (SouthernBiotech, Birmingham, AL, USA) secondary antibody in 1% BSA was added for 1 h at room temperature in the dark, washed twice with PBS, and stained with 0.1 μg/mL of 4′,6-diamidino-2-phenylindole (DAPI) for 5 min, followed by another wash. Finally, 50% glycerin (*w*/*v*, in PBS) was added, and the cells were visualized under Zeiss LSM710 inverted confocal microscope (Zeiss, Oberkochen, Germany).

For Western blot analysis, the treated cells were washed five times with PBS and lysed directly in the sodium dodecyl sulfate polyacrylamide gel electrophoresis loading buffer (Beyotime, Jiangsu, China). Each sample (10 μg of protein) was subjected to 8-16% polyacrylamide precast gels (Beyotime, Jiangsu,, China) and transferred to a polyvinylidene fluoride membrane (Bio-Rad, Hercules, CA, USA). Then the membranes were blocked with 5% milk and incubated with primary antibodies, overnight, at 4 °C. After being washed three times with PBST, horseradish peroxidase (HRP)-conjugated goat anti-rabbit IgG or HRP-conjugated goat anti-mouse IgG (SouthernBiotech, Birmingham, AL, USA) was added, and enhanced chemiluminescence Western blotting substrate (ECL; Millipore, Billerica, MA, USA) was used to visualize the bands. The primary antibodies, including rabbit anti-human ferritin (Product ab75973, Abcam, Cambridge, UK), rabbit anti-human solute carrier family 40 member 1 (SLC40A1) (Product ab235166, Abcam, Cambridge, UK), rabbit anti-human transferrin receptor (TFRC) (Product 13113, CST, Boston, MA, USA), rabbit anti-human nicotinamide adenine dinucleotide (phosphate) reduced:quinone oxidoreductase (NQO1), anti-heme oxygenase 1 (HMOX1) (Product GTX100235, GTX101147, GeneTex, Taiwan, China), and mouse anti-human glyceraldehyde-3-phosphate dehydrogenase (GAPDH) (SouthernBiotech, Birmingham, AL, USA) were used; furthermore, a Cell Cycle Regulation Antibody Sampler Kit (Product 9932, CST, Boston, MA, USA) was used for cell-cycle-related proteins Western blot analysis. All antibodies were diluted according to the manufacturer’s instructions. The intensity of the bands was quantified by using ImageJ software (v 1.50i).

### 2.11. Statistical Methods

Experiments were performed in biological/technical replicates, as indicated. Statistical significance was defined as indicated and specified in the figure and table captions. If not otherwise mentioned, significance was evaluated by GraphPad Prism 7 software (version 7.00), using one-way analysis of variance (ANOVA), and a *p*-value < 0.05 was considered statistically significant. Data were expressed as the mean ± standard deviation (SD) of at least three independent experiments. The size of nanoparticles samples was measured from TEM micrographs by using the software ImageJ (version 1.50i) and plotted with the SPSS (version 16.0).

## 3. Results

### 3.1. Particle Characterization

Membrane-enclosed MAGs from *Magnetospirillum gryphiswaldense* MSR-1 are single magnetic-domain cuboctahedral nanocrystals [29] that are typically organized one by one or arranged in a chain due to their membrane proteins [30,31]. Here, MAGs were harvested from cultured MSR-1 cells. The isolation buffer contained 0.5% Tween 20 to remove potential contaminating cellular materials that non-specifically bind to MAGs surfaces. TEM images of the particles revealed that isolated MAGs formed chain-like structures at low concentrations with mean core diameters of approximately 42 nm. Meanwhile, IONPs coated with amine were spherical, ~32 nm, and appeared to stick together (Figure 1A).

The negative staining TEM image of MAGs clearly shows the magnetite core was surrounded by an electron-light organic structure indicating the membrane (Figure 1B). Moreover, nearly all intrinsic MAG membrane proteins were identified by proteomic analysis of the isolated MAGs fraction (Appendix A). In addition to the MAG intrinsic membrane proteins, approximately 270 additional proteins were identified. However, in contrast to the identified 19 intrinsic MAG membrane proteins, which comprised only 6.4% of all identified proteins, yet accounted for 51.5% of the total protein abundance, the relative abundances of the additional proteins were much lower (Figure 1C) and likely represent impurities co-isolated with MAGs. Our results indicated that the lipid bilayer surrounding the magnetic core was preserved during MAGs isolation and that MAGs were intact and enveloped by a protein-rich biological membrane.

### 3.2. Cell Toxicity Analysis

We next assessed the cytotoxicity of isolated MAGs on ECs to determine the optimal dose for subsequent experiments. MAGs and IONPs were first incubated with plasma to form a protein corona to simulate the physiological environment. Protein amounts on the nanoparticles (containing equal Fe amounts) were measured by BCA assay. Fewer corona proteins were extracted from the MAG surface than the IONP surface (Appendix A). Furthermore, the zeta potential of MAGs slightly changed, which is consistent with our previous findings [22], which indicated that MAGs are shielded by membrane proteins, and that corona formation does not significantly alter the surface charge as the corona is also made up of proteins (Appendix A). For IONPs, the zeta potential after plasma incubation shifted from positive to negative, indicating that the corona covered the original IONP amine surface groups. The particle charge affects the internalization of nanoparticles by cells; therefore, corona attachment minimizes the potential charge-mediated effects and mimics the interactions that may occur physiologically [32,33].

After the initial period of growth to confluence, serum and growth factors were removed from the culture medium to maintain a single-layer cell culture during the treatment and avoid further passaging. ECs were exposed to 0, 10, 20, 40, or 70 μg Fe/mL of MAGs for 48 h, after which the medium was changed. As a comparison, cells were also treated with the same concentration of IONPs. Cell cycling was assessed after 48 h, and cell toxicity was determined on days 1, 4, 7, and 10. A flow cytometric analysis revealed that MAG-treated cells did not exhibit obvious changes in the cell-cycle distribution, while increasing the IONPs concentration caused a gradual decrease in the proportion of cells within the G2/M phase compared to the control group (Figure 2A).

Given that cell-cycle progression is partially dependent on the tightly regulated activity of cyclin-dependent kinases (CDKs), we analyzed the expression of cell-cycle regulators via Western blot and observed a slight decrease in CDK inhibitor p27 and p18 expression in the high-dose MAGs group (Figure 2B and Appendix A). Hence, MAGs also affected the cell-cycle checkpoint at a high dose; however, they did not impact the cell cycle overall. Meanwhile, mid-late G1 phase activators, such as cyclin D1/CDK4/6, were found to be downregulated, while the expression of the CDK inhibitor (p18) was upregulated in the IONPs group, indicating cell cycle arrest at G1. These results were concordant with the flow cytometry analysis.

Cell toxicity measurements showed that both MAGs and IONPs slightly promoted cell growth at an early stage (1 day). When the culture time increased, the proportion of living cells treated with MAGs at differential doses was about 100% compared with the control. In contrast, the proportion of living cells reduced after 1 day in the presence of high-dose IONPs, and the cell survival rate was reduced to 80% compared with the control at concentration of 70 μg Fe/mL (Figure 2C). These results were concordant with cell-cycle progression analysis, indicating that prolonged treatment with high levels of IONPs disrupted cell-cycle progression and inhibited cell growth. In contrast, MAGs had no apparent effect on cell cycle and a negligible influence on cell survival rate. Thus, compared with IONPs, MAGs exhibited weaker toxicity and were tolerated by ECs at a low-to-moderate concentration.

### 3.3. Cellular Internalization of MAGs and Iron Quantification

To assess the internalization and trafficking of MAGs by cells, we performed Prussian blue staining to track iron (Fe) localization, which also represents nanoparticle distribution, for up to 10 days in ECs. Within 2 h of initial incubation of ECs with MAGs, staining was observed along the cell border around the cell membrane. Subsequently, within 1 day, staining gradually accumulated in the cytosol near the nucleus. After 2 days, nearly all stained granules were internalized by the ECs. Meanwhile, in ECs incubated with IONPs, staining was observed along the cell border within the first 0.5 h and accumulated in the cytosol near the nucleus in the next 12 h (Figure 3A). Furthermore, following 10 days of incubation, with the media changed every two days, stained granules were still observed intracellularly, indicating the prolonged retention of MAGs and IONPs in ECs.

The TEM images showed that the internalized particles were first encapsulated in monolayer membrane vesicles which were commonly recognized as classical endosomes (Appendix A), and after 2 days, most of them appeared in the perinuclear vesicular structures (Figure 3B,C), presumably lysosomes or heterolysosomes which contained high protein content and numerous multilamellar bodies and appeared more contrasted [34]. Interestingly, IONPs were present as irregular agglomerated structures in the vesicles (Figure 3C), while some MAGs appeared to be more regularly arranged (Figure 3B). Cytosol and other intracellular compartments, including mitochondria, nucleus, and Golgi apparatus, were relatively free of nanoparticles.

Intact functional lysosomes possess an acidic internal pH (~4.5), which is required for optimal lysosomal hydrolase activity. A long-term analysis revealed that MAGs decomposed in vitro at a pH of 5.6 with proteases present [35]. Moreover, a 5-day culture suggested that minor degradation of the original mineral structures (MAGs from *M. blakemorei* strain MV-1) in a mammalian cell may occur [16]. For iron oxide nanoparticles, degradation in vivo and in vitro has been observed in many studies [34,35,36,37,38,39]. Hence, given that our study involved a 10-day treatment, both MAGs and IONPs may degrade into iron ions. Therefore, we analyzed the nanoparticles and released iron ions in ECs at different timepoints, using ICP-MS (Figure 4A). The intracellular particle Fe levels increased in IONP-treated cells earlier than in MAG-treated cells. More specifically, the concentration reached a maximum after 12 h for the IONPs group, while a similar concentration was not achieved in the MAGs group until day 2. During the following 8 days, the concentration remained relatively constant, which is consistent with the Prussian blue staining (Figure 3A). Meanwhile, the curves representing the level of released iron ions in cells upon treatment with the two types of nanoparticles were relatively smooth within the first 2 days. However, after 2 days in MAG-treated ECs, the intracellular iron ion levels increased significantly and kept increasing until the end of the observation period, indicating a relatively high rate of MAG iron core dissolution. In contrast, the iron ion released from IONP-treated cells showed a slight increase after 2 days, indicating a slow rate of IONP dissolution.

Collectively, our analyses demonstrate that, although IONPs and MAGs both contain surface amino groups, MAGs exhibited unique properties, including fewer corona proteins, relatively weaker toxicity at a high-dose treatment (70 μg Fe /mL), slower internalization, and rapid dissolution.

### 3.4. Overview of Differentially Expressed Proteins in MAG-Treated ECs

Previous studies show that MAGs’ degradation into remnants in vitro takes dozens of days [35], and they are retained in mice for several weeks [40]. Our ICP-MS results (Figure 4A) demonstrated continuous iron ion dissociation in the cell. However, the MAGs did not exert significant effects on cell survival or morphology over a prolonged exposure time in our study. Thus, MAGs and their by-products (iron ion) must have been handled by intracellular mechanisms of ECs. To characterize the mechanisms associated with ECs’ response to MAGs, particularly related to the released iron ions, a TMT-labeling quantitative proteomics analysis was performed. We detected expression variations of proteins in MAG-treated ECs by comparing them with the control ECs without MAGs treatment. A total of ~8520 proteins were identified and quantified, and 6859 proteins were shared (Appendix A), revealing that the majority of the proteins was identified at each timepoint. To make the quantitative results more robust, for each timepoint, we set the criteria for downregulation to a protein with ratio ≤ 0.83 in at least two biological repeats, with its average ratio (Avg.) ≤ 0.83 and *p*-value < 0.05, and upregulation was determined with a ratio of ≥1.2 in at least two biological repeats, with its average ratio (Avg.) ≥ 1.2 and *p*-value < 0.05. After the pruning procedure, a total of 158 differentially expressed proteins were identified (Appendix A). Figure 4B provides an overview of scatter plots, which showed the differential expressed proteins in each timepoint. The obtained results evidence that MAGs induce perturbations on intracellular pathways.

GO and pathway analyses were performed to determine the enriched cellular components, molecular functions, and pathways associated with the differentially expressed proteins in MAG-stimulated cells. Statistical tests were performed to assess the enriched categories with reference to the KEGG pathways, GO, WikiPathways, and REACTOME databases. Appendix A summarizes the enriched cellular component or functions and pathways with *p* < 0.05, using the above four databases for proteins’ regulated upon MAGs treatment at four timepoints (details are shown in Appendix A). The cellular locations of differentially expressed proteins were significantly enriched in endosomes and lysosomes (Figure 5A), while enriched functions or pathways were roughly classified into four categories: lysosome- and vesicle-related, iron-ion homeostasis, membrane-receptor-related uptake, and cell proliferation (Appendix A). As mentioned above, most of the internalized MAGs were sequestered in lysosome-like vesicles. Here, the differentially expressed proteins were further enriched in regard to lysosome function and iron metabolism, indicating the underlying mechanisms partially related to ECs to handle MAGs and their by-products (Figure 5B).

### 3.5. MAGs Induce Changes in Endosomal–Lysosomal Proteins

Through quantitative proteomics, we identified a group of proteins associated with vesicle biogenesis and lysosome function (Figure 5). Specifically, H(+)/Cl(−) exchange transporter 5 (CLCN5), a proton-coupled chloride transporter which is associated with normal acidification of the endosome lumen and is important for the recycling of the receptor back to the apical membrane for further endocytosis [41], as well as proteins of the AP complex family (AP-1 complex subunit beta-1 [AP1B1] and AP-4 complex subunit mu-1 [AP4M1]), which mediate both the recruitment of clathrin to membranes and protein sorting in the endosomal-lysosomal system [42], were upregulated. Similarly, lysosomal enzymes, including acid hydrolases, such as protease (legumain, LGMN), lipase (lysosomal acid lipase/cholesteryl ester hydrolase, LIPA), and glycosidase (heparanase, HPSE), as well as other enzymes, such as prosaposin (PSAP) and palmitoyl-protein thioesterase 1 (PPT1), which participate in the lysosomal degradation of lipids and lipid-modified proteins, were upregulated following MAGs treatment. Hence, endosome- and lysosome-related processes were affected by MAGs, and the differentially expressed proteins may contribute to MAG-carrying vesicle transportation, as well as lysosomal degradation of the MAG membrane and its attached corona proteins.

### 3.6. Iron Metabolism Regulation in ECs

As discussed above, an increase in soluble forms of iron was observed after MAGs uptake, and the proteomics analysis revealed that processes associated with iron ion homeostasis, including iron metabolism, iron uptake and transport, and ferroptosis, were affected. Normal iron homeostasis is mediated through iron absorption, utilization, secretion, or intracellular deposition executed by cells [43]. Usually, iron absorption occurs through iron binding to transferrins in the serum, which is recognized by the TFRC on the cell membrane. Alternatively, lactotransferrin (LTF) provides additional sources of iron absorption through different uptake pathways on the cell membrane [44]. After iron absorption, membrane protein natural resistance-associated macrophage protein 2 (SLC11A2), which is important in metal transport, mediates iron release from the endosome into the cytoplasm [45]. The released iron will join the intracellular labile iron pool, where it can be used for the metabolic function of the cell or stored in the form of ferritin. Ferritin components are iron-storage proteins that comprise ferritin heavy chain (FTH1) and ferritin light chain (FTL), which allow subsequent iron entry into the ferritin mineral core and have protective effects on ferroptosis [46]. Iron is mainly exported by SLC40A1 on the cell membrane, but it can also be exported as ferritin through exosomes [47]. Cellular iron homeostasis is maintained by the iron regulatory proteins, which function as iron sensors, such as iron-responsive element-binding protein 2 (IREB2) and cytoplasmic aconitate hydratase, at the translational level. Iron regulatory proteins bind to iron-responsive elements and translationally regulate iron metabolism-related proteins (for example, TFRC, SLC11A2, SLC40A1, FTH1, and FTL), control intracellular iron levels, and thus regulate cell growth and ferroptosis [43].

Here, our proteomics results showed that, under MAGs treatment, the expression of IREB2 was initially decreased and subsequently returned to normal (Figure 5 and Appendix A). Previous studies showed that IREB2 is mainly regulated via protein degradation and is degraded when iron is in excess [46,48,49]. Concordantly, the expression of ubiquitin-60S ribosomal protein L40 (UBA52)—involved in IREB2 ubiquitylation for proteasomal degradation—was upregulated in ECs upon initial treatment with MAGs (Figure 5 and Appendix A). Meanwhile, the abundance of TFRC, LTF, and SLC11A2 decreased, whereas that of FTH1 and FTL was markedly elevated (Figure 5 and Appendix A). The abundances of FTH1 and FTL were confirmed by immunofluorescence and that of TFRC, FTH1, and FTL was confirmed by Western blotting analysis (Figure 6A). The immunofluorescence signal for ferritin (FTH1 and FTL) in the control ECs was weak and randomly dispersed throughout the cytosol (Figure 6B). In comparison, at 10 days post MAG treatment, ferritin staining was strong and clustered. Although SLC40A1 was not identified in the proteomics analysis, through Western blot, we determined that its abundance was increased upon MAGs treatment at days 7 and 10 (Figure 6A and Appendix A). Thus, our results revealed that MAGs may induce IREB2 degradation, which subsequently regulate genes associated with iron metabolism, including the translational inhibition of genes involved in iron import (TFRC, LTF, and SLC11A2) and translational induction of genes involved in iron storage (FTH1 and FTL) and export (SLC40A1).

### 3.7. Cell Oxidative Stress Analysis

Iron is vital for almost all living organisms, as it participates in a wide variety of metabolic processes, including oxygen transport, DNA synthesis, and electron transport [49]. The physiological level of iron promotes cell growth; excess free iron may affect cell death including ferroptosis, which relies on free iron accumulation and facilitates oxidative damage of lipid peroxidation and membrane oxidative damage through either the production of highly reactive hydroxyl free radicals in the Fenton reaction or the activation of Fe-containing enzymes [44,46]. MAGs treatment could increase intracellular iron ion level and induce iron homeostasis regulation. Here, we measured the lipid peroxidation, protein oxidation, and expression of certain cytoprotective proteins to verify whether the iron ion released following MAGs treatment is managed by cell-iron-homeostasis regulation.

Oxidative damage induced by MAGs in ECs was assessed by quantifying the relative levels of MDA, a marker of lipid peroxidation, and carbonyl groups resulting from protein oxidation. MAGs induced slight lipid peroxidation and protein oxidation after 4 days in ECs, at 10 μg Fe/mL (Figure 6C). However, the relative levels of MDA and carbonyl groups did not significantly increase with increasing exposure time. Upon longer MAGs exposure (4 days to 7 or 10 days), the MDA level increased by approximately 1.4-fold compared to the control, while protein carbonyl content was ~1.2 fold higher than in the control. In addition, the expression of cytoprotective proteins, such as HMOX1, which involved in limiting oxidative damage in ferroptosis, and reactive oxygen species (ROS) detoxification enzymes, such as NQO1 [50], did not exhibit significant changes upon MAGs treatment (Figure 6B), thus suggesting the absence of significant oxidative stress in ECs throughout the 10-day exposure period. These results suggest that the intracellular iron balance was controlled, and the elevated intracellular iron concentration induced by MAGs degradation was managed by regulators associated with iron metabolism, without inducing significant oxidative damage.

## 4. Discussion

MAGs represent biogenic, magnetic nanoparticles that are biosynthesized by magnetotactic bacteria [12]. The core of MAGs from MSR-1 cells is primarily composed of Fe_3_O_4_ [23], which is similar to that of magnetic iron oxide nanoparticles; however, MAGs are enveloped by a proteinaceous phospholipid bilayer. Moreover, their high crystallinity, strong magnetization, anisotropic shape, and potential application for spatial magnetic guidance have made them a research hotspot [7]. Additionally, their enveloped protein-rich membrane resembles the bioinspired core-shell nanoparticles [51]. Therefore, MAGs are peculiar magnetic iron oxide nanoparticles that differ from synthetic magnetic iron oxide nanoparticles with a chemically modified surface. Various applications have been proposed for the use of MAGs in different biomedical fields, including tissue engineering, cell tracking, tissue contrast, and tumor imaging and treatment [52,53,54]. However, compared with artificial magnetic iron oxide nanoparticles, studies about the MAGs’ action in cells are relatively limited. The question arises as to what will happen when MAGs enter a cell: will MAGs be eliminated or degraded, and if so, how will cells handle the degradation by-products? In the present study, we investigated MAG-induced effects in normal vascular ECs, as ECs comprise the luminal lining of all vascular system components and commonly interact with nanoparticles [20]. Furthermore, ECs are responsible for the transport of nanoparticles into solid tumors [21]. Previous studies examining MAGs uptake by cultured HepG2, ARPE-19, Raw 264.7, FaDu, and HUVEC cell lines employed short exposure times (~72 h) [12,13,15,22]. Unlike immortalized and tumor cell lines, which require passaging and cannot typically be maintained for long in culture without serum, normal vascular ECs can maintain a single-layer culture without serum after cells were grown to confluence in ~10 days. Under these conditions, we minimized the impact of cell division on nanoparticle dilution and enabled a longer-term in vitro observation.

The isolated MAGs were examined by TEM, and the membrane was clearly visible. Previous proteomic approaches indicated that Mam and Mms proteins are present in the MAG membrane [55]. This agrees with our LC-MS/MS analysis, which identified Mam and Mms proteins as the most abundant proteins in the MAG membrane proteome. These results demonstrate that the isolated MAGs have an integrated structure with an inorganic core and a biogenic protein-rich membrane.

To assess the cytotoxic effects of isolated MAGs over 10 days, we applied concentrations (0-70 μg Fe/mL) within the dosage range administered in contrast agents for clinical diagnostic protocols [33,56], in which the administered Fe concentration in the blood varied from about 3 to 75 μg/mL [57,58]. IONPs with a similar particle size were assessed for comparison. The cell-cycle analysis suggested that, in certain concentrations, IONPs could cause cell-cycle arrest in a dose-dependent manner. Similarly, Fe_3_O_4_ nanoparticles could cause significant changes in the cell-cycle phase distribution in other cell lines [59,60]. In contrast, only the highest dose of MAG treatment caused a slight decrease in checkpoint proteins p27 and p18; however, it did not impact the cell-cycle overall. Furthermore, cell-vitality data confirmed the biocompatibility of MAG concentrations up to 70 µg Fe/mL, which is in accordance with values reported for other cell lines [16]. In comparison, IONPs decreased cell viability after high-dose treatment. Hence, compared with IONPs, MAGs were found to be better tolerated by ECs at high-dose treatment. Indeed, the induction of low/negligible cytotoxicity is a requisite for application of nanoparticle intracellular long-term fate analysis to ensure that the cells remain alive long enough for sufficient analysis.

MAGs and other iron oxide nanoparticles have been found in classical endosomes and lysosomes [12,15,61,62]. Our ultrathin sections (50 nm) of ECs following incubation with MAGs and IONPs also revealed a large number of internalized particles in endosome- and lysosome-like vesicles, as is consistent with observations in other cell lineages. However, the two nanoparticles entered cells at different rates, and MAGs arranged regularly in vesicles. The interaction of nanoparticles with cells is initiated by their binding to the cellular receptors or membranes followed by their uptake. It was reported that the interaction can be affected by the protein corona [62]. In HeLa cells, the MAG membrane was retained in MAGs located within endosomes [16]. Hence, we assumed that the different cellular internalization rates between MAGs and IONPs may relate to their protein corona, and the regular arrangement of MAGs in ECs may be due to the presence of the membrane proteins, which allowed the MAGs to be arranged in lines in MSR-1 cells and did not fully break down in 48 h. However, further detailed studies are necessary to address these questions.

Intact functional lysosomes possess an acidic internal pH (~4.5), which is sufficient to dissolve iron oxide nanoparticles [38]. Compared with the internalized iron particles, the amount of released iron ion is relatively low. ICP-MS analysis provided a sensitive assessment of the trace release. Compared with IONPs, MAGs exhibited faster iron ion release. This suggests that a small fraction of the MAGs were degraded. Concordantly, previous studies suggest minor degradation of MAG (isolated from *M. blakemorei* strain MV-1) crystals in HeLa cells [16], and in vitro MAGs are more readily decomposed than spherical Fe_3_O_4_ caped with amino silane (10 nm) [35]. These differences in degradation profiles may be related to particle surface modifications, as MAGs are covered by a biosynthesis membrane and proteins that are easily degraded in the lysosome. In addition, it has been demonstrated that, in the lysosome, positively charged particles can function as weak proton sponges, thereby increasing the local pH around the nanoparticles, leading to a slower dissolution rate [52]. Hence, we suspect that, after decomposition of the plasma corona, the positively charged surface of IONPs will lower the cores’ dissolving rate. However, further detailed investigation is warranted to elucidate the degradation mechanism of MAGs.

Recent studies have reported the signaling pathways affected by different iron-based nanoparticles, including inflammation, oxidative stress, autophagy, and cell-cycle arrest [62], and iron-based nanoparticles easily join the physiological iron pool and become processed via iron homeostasis regulatory mechanisms [36,62]. However, few studies have reported that MAGs impacted molecular regulation upon long-term treatment. Our results show that the cellular components of differentially expressed proteins were significantly enriched in endosomes and lysosomes. Among the upregulated proteins, CLCN5, AP1B1, and AP4M1 mediate clathrin receptor recycling. Considering that endocytosis is the main route by which nanoparticles enter cells [62], and clathrin-mediated endocytosis has important roles in the cellular uptake of MAGs (isolated from *Magnetospirillum magneticum* AMB-1) by HepG2 cells [15], the upregulated expression of these proteins indicate the intracellular vesicles which contained MAGs consume the cell-surface receptors and induce ECs to accelerate the process of receptor recycling. Furthermore, the increased lysosomal enzymes associated with protein and lipid degradation suggest that cells tend to trigger/accelerate MAG degradation in lysosomes.

Along with MAG degradation, iron-ion release occurred, and the enriched cellular process indicates that the specialized metabolic process that regulates iron in the organism also handles MAG products. Remarkably, the levels of ferritin, an iron storage protein [46], increased significantly in ECs and tended to cluster as bright dots in fluorescence staining at 10 days post MAGs treatment. Ferritin regulates the biotransformation of iron oxide nanoparticles by binding to the released iron; the iron-rich ferritins are then either dispersed in the cytoplasm or form large assemblies [34,63,64,65]. Hence, the increased expression and clustering of ferritin suggests an ongoing transformation process of MAGs, by which the labile iron species, resulting from particle dissolution, could be stored by another nontoxic endogenous storage protein. Moreover, these findings suggest that ECs can remodel MAGs particles. Besides the iron-storage proteins, the expressions of iron-sensor protein IREB2, protein UBA52, which involves in IREB2 proteasomal degradation [66], iron-import proteins TFRC and LTF [44], iron-transport protein SLC11A2 [45], and iron-export protein SLC40A1 [47] were impacted. Hence, the elevated intracellular iron-ion level induced by MAGs degradation was sensed by ECs, which led to certain signaling events, including IREB2 degradation, the downregulation of iron-import proteins to limit iron absorption, the downregulation of iron-transport proteins to slow release of iron ions into the cytoplasm, and the upregulation of iron-storage proteins and iron exporters to control the intracellular iron balance. The activation of the iron regulatory pathway is consistent with MAGs degradation. Such a mechanism may offer a chance for the ECs to regulate the labile iron pool generated by the degradation products of MAGs, thus minimizing toxicity and oxidative damage.

Overall, our proteomics results present a general picture of changes inside ECs upon MAGs treatment at the molecular level. Although variations at the cellular level were not apparently observed under low-concentration treatment conditions, various regulations were observed at the molecular level. These alterations at the protein level and the potential for the regulation of signaling pathways help to elucidate the response of ECs to MAGs and provide insights into the potential biological effects of MAGs. However, cellular signal regulation is a highly complex network; therefore, more efforts are required to further explore the functions of these molecules in cells and allow us to better understand fundamental signaling modulations induced by MAGs, as this is essential to overcome in vivo application challenges.

## 5. Conclusions

In this work, we carried out a comprehensive study of the biological effects of isolated MAGs on ECs at the cellular and molecular level. Compared with those treated with IONPs, MAG-treated ECs do not cause a significant change in cell-cycle distribution and maintain normal cell survival rates for up to 10 days. Following uptake by ECs, MAGs primarily localize in lysosome-like vesicles. Prolonged observations indicate that iron ions are released into the cell, indicating slight degradation of the particles. Through quantitative proteomics, we found that MAGs can induce the differential expression of many proteins and affect cellular processes related to the lysosomal digestive function and iron coping mechanisms. Hence, our findings demonstrate that MAG degradation products can be incorporated into an organism’s iron metabolism pathway, thus preventing an uncontrolled increase in uncomplexed iron to generate excessive ROS. Considering that MAG permits detection with clinical MRI devices even at very low concentrations [67,68], the combination of cellular- and molecular-level analyses confirms that MAGs are biocompatible and should have excellent potential for cellular biotechnology and biomedicine applications.

## Figures and Tables

**Figure 1 nanomaterials-12-03995-f001:**
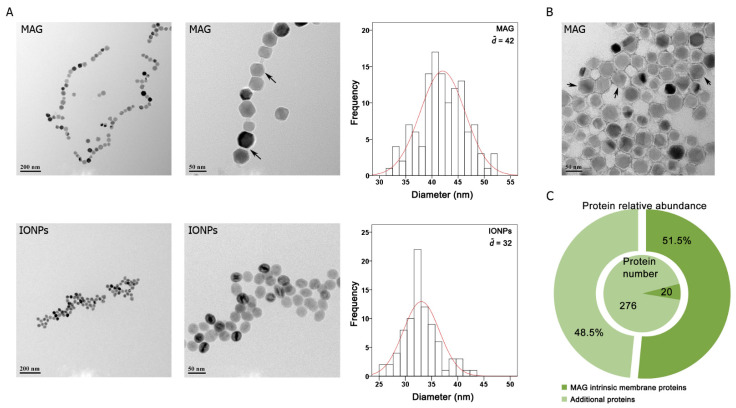
Characterization of nanoparticles. (**A**) TEM images of typical magnetosomes (MAGs) and amine magnetic iron oxide nanoparticles (IONPs). Arrows indicate MAG membrane. Frequency distributions show the particles size distribution of typical MAGs (~42 nm in diameter) and IONPs (~32 nm in diameter), as determined by TEM. (**B**) Negative staining TEM image of typical MAGs. The magnetite core is surrounded by an electron-light organic shell (indicated by black arrows), representing the membrane. (**C**) Proteomic analysis of MAGs protein composition and relative abundances. The identified proteins were classified into two groups (MAG intrinsic membrane proteins and additional proteins. Details are given in Appendix A).

**Figure 2 nanomaterials-12-03995-f002:**
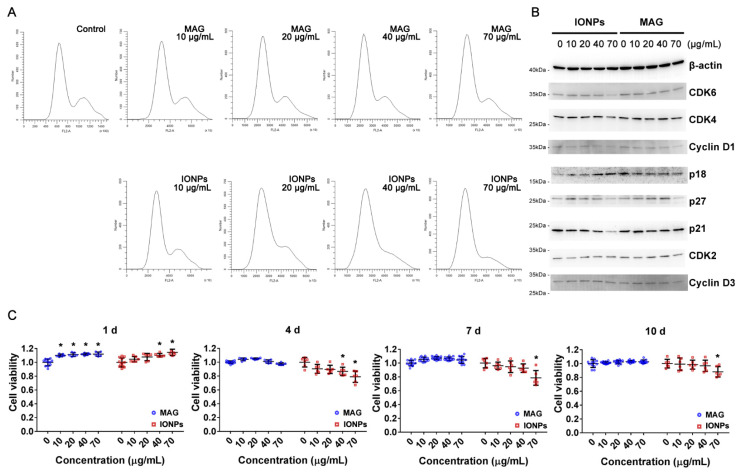
Analysis of cell cycle and cell viability of endothelial cells (ECs) treated with or without nanoparticles. (**A**) Cell-cycle distribution of ECs after exposure to MAGs or IONPs for 48 h over a dose range of 10-70 μg Fe/mL. Control, ECs without nanoparticle treatment. (**B**) The expressions of proteins that are essential for cell-cycle progression in ECs treated with MAGs or IONPs for 48 h over a dose range of 0-70 μg Fe/mL. β-actin, loading control; CDK, cyclin-dependent kinase. (**C**) ECs viability analysis in the presence of MAGs or IONPs or not. The treated samples were normalized to control cells (treated with 0 μg Fe/mL particles) for each timepoint, respectively, and the viability of control cells was regarded as 1.0 (n ≥ 6, data are shown as mean ± SD; * *p*-value < 0.05; d, day).

**Figure 3 nanomaterials-12-03995-f003:**
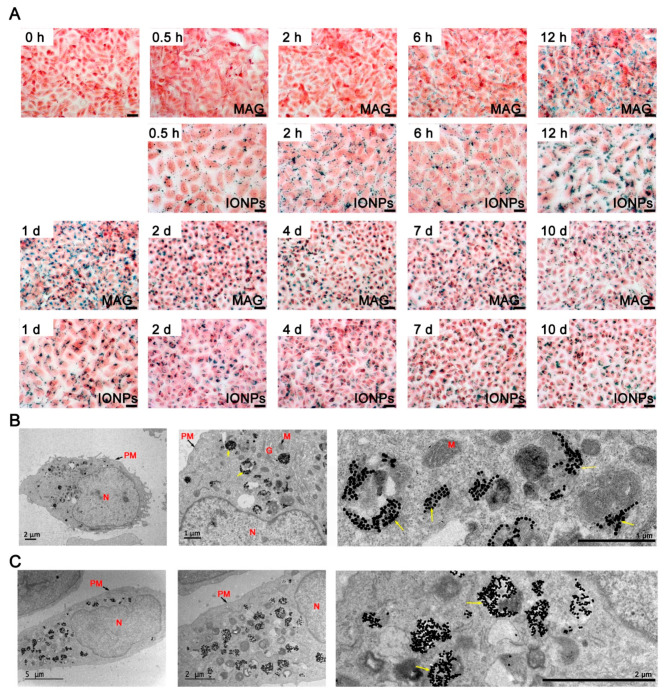
The cellular internalization of nanoparticles in ECs. (**A**) Optical microscopy Prussian blue stained images showing the internalization of MAGs and IONPs by ECs at different timepoints. ECs were treated with MAGs or IONPs at 10 μg Fe/mL. The blue dots representing the Fe and the cells were stained by eosin staining. Scale bar, 20 μm. (**B**) TEM image of MAGs (yellow arrows) within vesicle-like structures in the cytoplasmic region of ECs after 48 h incubation (10 μg Fe/mL). (**C**) TEM image of IONPs (yellow arrows) within vesicle-like structures in the cytoplasmic region of ECs after 48 h incubation (10 μg Fe/mL). Abbreviations: PM, plasma membrane; N, nucleus; M, mitochondria; G, Golgi apparatus.

**Figure 4 nanomaterials-12-03995-f004:**
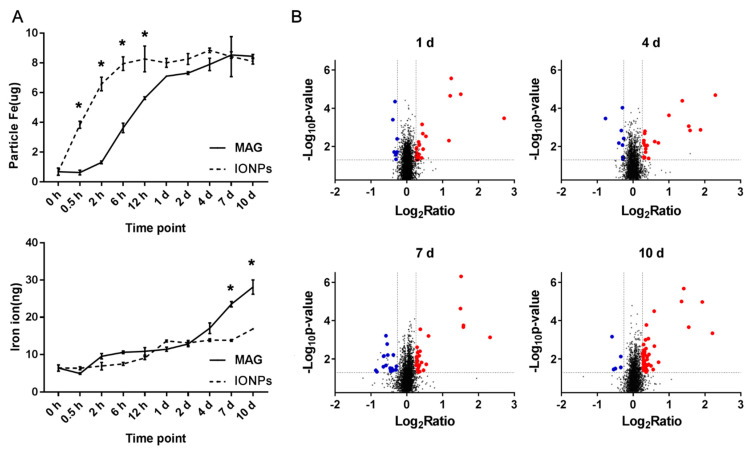
Intracellular iron concentrations of ECs treated with nanoparticles and overview of the identified proteins in ECs treated with MAGs. (**A**) Inductively coupled plasma-mass spectrometry (ICP-MS) analysis of insoluble iron particles (particle Fe) and soluble iron (iron ion) amount at different timepoints in ECs treated with MAGs or IONPs (10 μg Fe/mL). Data are shown as mean ± SD, n = 3 for each timepoint, and * *p*-value < 0.05 means the amount of insoluble iron particles or released iron ion is significantly different between IONPs and MAGs treated ECs. (**B**) Scatter plots of quantitative proteomics results of ECs treated with MAGs compared to the control. Blue and red dots represent the decreased and increased proteins in ECs at days 1, 4, 7, and 10, respectively. The ratio represents the average ratio of each protein quantified from three biological replicates. Control, ECs without MAGs treatment.

**Figure 5 nanomaterials-12-03995-f005:**
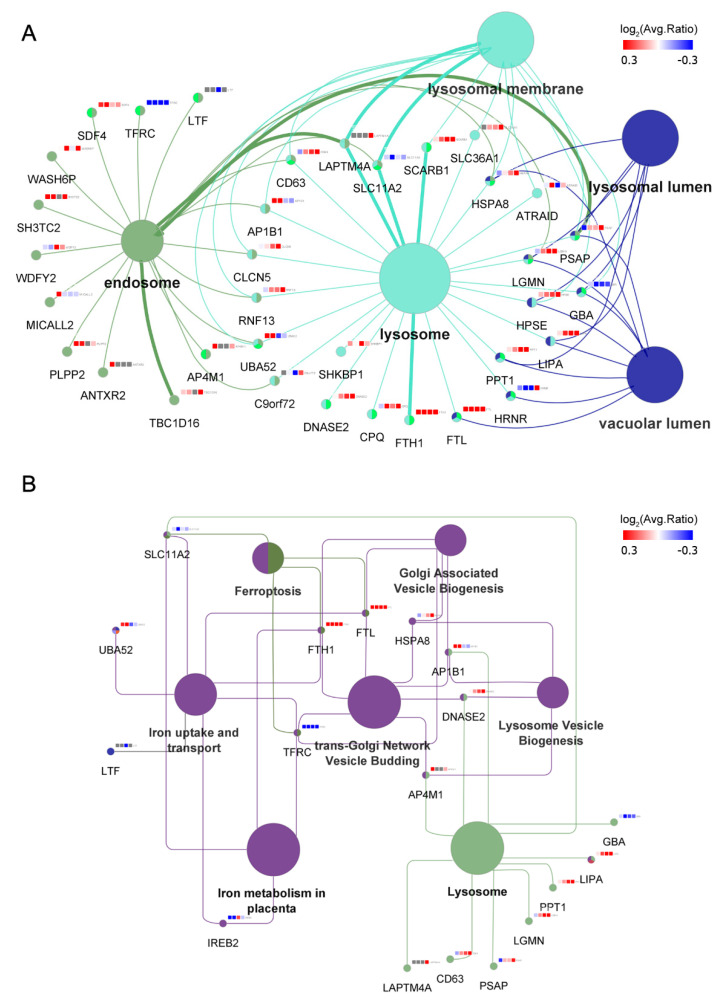
Bioinformatic enrichment analyses of differentially expressed proteins of MAG-treated ECs. Each protein is represented by a gene name, with the superscript color on gene name showing the protein expression ratio at days 1, 4, 7, and 10 (red, upregulated; blue, downregulated). (**A**) Network showing the enriched cellular component terms (*p*-value < 0.05) related to endosome and lysosome. Enriched terms were obtained from the GO Cellular Component database. The *p*-value and details of every enriched term are listed in Appendix A. (**B**) Network showing the enriched function and pathway terms (*p*-value < 0.05) related to iron metabolism and lysosome. Enriched terms were obtained from four databases (KEGG pathways, GO Molecular Function, Wiki Pathways, and REACTOME Pathways). The *p*-value and details of every enriched term are listed in Appendix A.

**Figure 6 nanomaterials-12-03995-f006:**
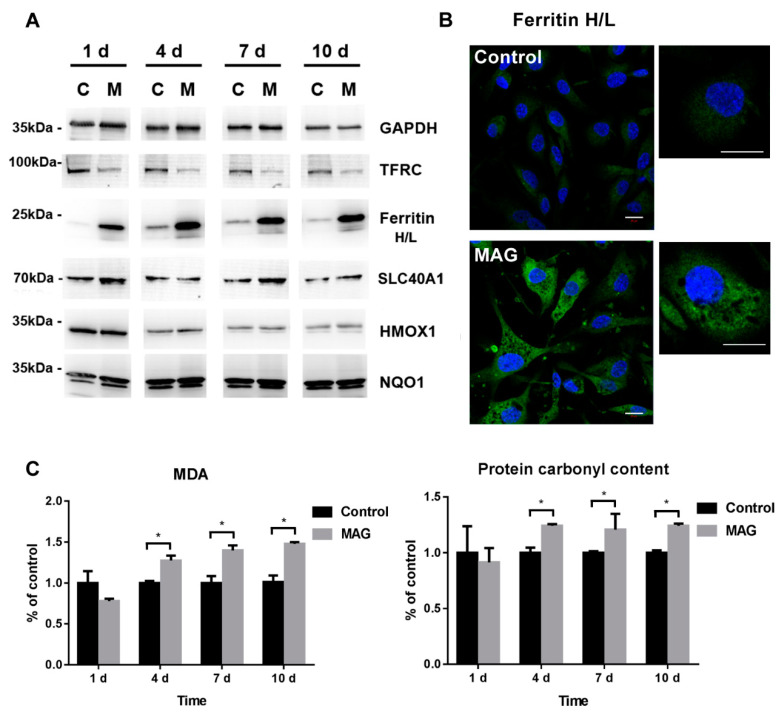
Evaluation of protein expressions and oxidative damage induced by MAGs in ECs. (**A**) The expression levels of proteins that are essential for limiting oxidative damage and iron metabolism in ECs. ECs were incubated with or without MAGs (10 μg Fe/mL) and protein expressions were analyzed via Western blot on days 1, 4, 7, and 10. C, lysis of control cell without MAGs treatment; M, MAG-treated cell lysis. (**B**) Immunofluorescence observation of ferritin (green) expression in ECs treated with or without MAGs (10 μg Fe/mL) on day 10. Cell nuclei (blue) were stained with 4′,6-diamidino-2-phenylindole (DAPI). Scale bar: 20 μm. (**C**) Relative levels of malondialdehyde (MDA) and protein carbonyl groups in ECs incubated with or without MAGs (10 μg Fe/mL). Lipid peroxidation marker, MDA, and carbonyl groups resulting from protein oxidation were measured by the MDA Assay Kit and Protein Carbonyl Colorimetric Assay Kit at different timepoints, as indicated. The treated samples were normalized to control at each timepoint, respectively, and the control was regarded as 1.0 (n > 3 for each timepoint, data are shown as the mean ± SD, * *p*-value < 0.05, control, ECs without MAGs treatment; MAG, ECs treated with MAGs).

## Data Availability

The data presented in this study are available upon request from the corresponding author.

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
