# Peer review of "Bacterial Magnetosomes Release Iron Ions and Induce Regulation of Iron Homeostasis in Endothelial Cells"

_nanomaterials, 2022, doi:10.3390/nano12223995_

Round 1
Reviewer 1 Report
Manuscript ID: 1990960
Author list
Wenjia Lai
Dan Li
Qingsong Wang
Yan Ma
Jiesheng Tian
Qiaojun Fang
“Bacterial Magnetosomes Release Iron Ions and Induce Iron Homeostasis Regulation in Endothelial Cells”
SUMMARY
The manuscript reports various features of endothelial cells (ECs) that have taken up iron nanoparticles: either isolated magnetosomes (MAG) from magnetotactic bacteria or chemically synthesized iron oxide nanoparticles (IONP). Both MAG and IONP appear to be compartmentalized within ECs and are not dispersed in the cytoplasmic compartment. A 10 day time course shows different rates of nanoparticle uptake and very little difference in cell viability or toxicity in the presence of this extra iron load. Most of the study then focusses on characterizing MAG in ECs to better understand how MAG affect cellular iron homeostasis. An extensive proteomic analysis is presented that suggests lysosomal involvement in MAG sequestration and appropriate changes in iron metabolism in response to an increase in soluble forms of iron.
GENERAL COMMENTS
The topic is interesting and timely, considering the broad applications contemplated for bacterial magnetosomes in mammalian cell tracking. However, the authors have a tendency to over speculate when drawing conclusions from the presented experiments. Keep in mind that lysosomes containing MAG were not isolated in this work. Suggestions have been made to guard against over interpretation of the data, which otherwise stands alone as a good study describing changes in mammalian cells induced by exogenous iron nanoparticles from magnetotactic bacteria.
Throughout the manuscript, there are grammatical points that need addressing. Refer to the text underlined in green within the PDF for areas needing revision, and please ignore the yellow highlighted text which was for review purposes only.
TITLE
I suggest a minor modification to avoid a long string of adjectives.
“Bacterial Magnetosomes Release Iron Ions and Induce Regulation of Iron Homeostasis in Endothelial Cells”
ABSTRACT
On line 23, what is meant by a “natural-natural interface”? Define this term.
On line 29, correct the grammar and order of ideas.
“Magnetosomes were retained in cells and found to gradually gather in the lysosomal compartment.”
INTRODUCTION
On line 47, distinguish bacterial from mammalian cell membranes when referring to magnetosomes.
“... are naturally covered by a protein-rich phospholipid bilayer, homologous to bacterial cell membranes.”
In the paragraph beginning on line 58, ensure that bacterial cells are distinguished from mammalian cells.
On line 61, do you mean toxicity of MAG has been examined in different non-magnetic bacterial and mammalian cell types?
On lines 67-68, are you referring to any host cell? non-magnetic cells? mammalian cells?
On line 58, correct the phrase.
“The broad potential application of MAG in a myriad of fields ...”
On line 65, correct the grammar and be sure that “natural-natural interface” is defined. (Refer to the comment in the Abstract.)
“As a naturally synthesized nanoparticle ...”
In the last paragraph of the Introduction beginning on line 70, give more context about the animal model(s). “Following injection” into what?
On line 75, are you referring to the application of MAG in cancer models? If yes, then specify this.
On line 78, correct the grammar.
“... form protein corona after interaction with ... and uptake by ...”
On line 82, correct the grammar.
“We show (here) that MAG are transported to lysosomes and retained by cells throughout the time course.”
On lines 83-84, correct the grammar. It is understood that enzymes are proteins.
“The MAG induce ECs to up-regulate vesicle biogenesis and expression of lysosomal enzymes.”
On lines 85-86, correct the grammar and smooth the text as follows.
“... the released iron ... subsequently triggers the regulation of ECs iron homeostasis ...”
On line 87, “at low dose” of what?
On lines 88-89, correct awkward wording.
“... and maintain homeostasis, indicating that MAG are biocompatible as well as useful for ...?” Finish the thought.
On line 90, what “real-world” applications are you referring to?
METHODS
The title of section 2.1 should read “Bacterial strain and magnetosome preparation”.
On line 99, did you mean spectrophotometry?
On line 101 of section 2.2, what is SHA-30?
On line 104, correct the grammar.
“Negative staining was performed ...”
On line 110, smooth the wording as follows.
“The protein content of MAG with or without ...”
Address the run-on sentence on lines 123-126 in section 2.3.
On line 124, place a period after PBS. The next sentence should begin with “MAG or IONPs with corona were suspended in conditioned medium ...”
On line 126, how much serum was added to cultured cells?
On line 126, use the correct scientific notation for water.
On line 140 in section 2.4, clarify what the control sample is.
The title of section 2.5 should read “Flow cytometry analysis of the cell cycle”.
On line 151 in section 2.6, define MDA.
On line 158, use the following wording.
“Protein oxidation level was estimated by measuring protein carbonyl content using ...”
On line 184 in section 2.7, it sounds like you are referring to soluble versus insoluble iron. Consider the following suggested revision.
“The amount of soluble iron and insoluble iron particles in cells was measured using ICP-MS.
In general, section 2.8 requires editorial revision.
On lines 193 and 199, define all abbreviations.
On line 198, the correct term is “technical repeats”.
In the document describing supporting information, including Figure S1, define all abbreviations. The general science readership should be able to follow your protocol(s).
On lines 211-212 in section 2.10, what is the difference between PBS and 0.1 M PBS?
For the benefit of all readers, indicate which paragraph corresponds to immunofluorescence and which corresponds to immunoblotting.
On line 215, how much rabbit anti-ferritin was used?
On line 216, did you mean FITC-conjugated goat anti-rabbit immunoglobulin? The secondary must recognize the primary antibody.
On lines 218 and 227, define all abbreviations.
On lines 228-234, indicate how much primary antibody was used for each blot.
On line 239 in section 2.11, define all abbreviations.
RESULTS
On line 280, clearly indicate that both types of nanoparticle were incubated with plasma.
On lines 283-284, provide a reference to corroborate previous findings.
On line 294, the control is described as IONP treatment; however, the figure reports another control. Please clarify.
For Figure 2A, why not integrate the area under the curve to quantify changes in the distribution of cells?
Lines 309 to 317 are a repeat of the previous paragraph.
In the paragraph beginning on line 318, why not present densitometric analysis of the western blot results?
On lines 333-337, a case is made for the toxicity of IONP relative to MAG. Do any of the data in Figure 2C stand out as significantly different? If not, reword the description of results to this effect.
On line 341, correct the word choice.
“Within 2 hours of initial incubation of ECs with MAG, staining was observed along the cell border near the membrane.”
On line 342, clarify the description of Prussian blue staining. Was Prussian Blue initially observed on both sides of the plasma membrane?
How do you know the TEM structures are lysosomes? Perhaps the most interesting feature is that all particles, whether MAG or IONP, were compartmentalized in a vesicle in these mammalian cells. Maybe this is why the cellular toxicity was not much different between each form of iron particle; the endothelial cells have sequestered the iron particles to minimize interactions with other cellular components.
Beginning on line 368, take care to distinguish bacterial turnover of MAG from mammalian cell handling of these foreign particles.
On lines 376-378, is the difference between the two curves significant from days 1-10?
On line 399, I’m not sure that there are any significant differences in toxicity between MAG and IONP. Use statistical analyses to be convincing.
On line 401, rephrase the sentence to avoid unfounded judgement.
“Decomposition of MAG has been studied in vitro (ref) and in vivo (ref) and may influence how easily these nanoparticles are eliminated from the body. Our ICP-MS results (Figure 4A) demonstrated ...”
The paragraph beginning on line 407 is poorly expressed. Seek editorial assistance. It could be combined with the preceding paragraph.
Can you relate MAG-induced changes in protein expression to IONP-induced changes in protein expression? Remind the reader how the control reference ECs were treated.
On lines 430 to 431, avoid making conclusive statements from partially characterized results. The data are consistent with the notion that MAG are sequestered in lysosome-like vesicles. Is ferrous or ferric iron exported from lysosomes?
On line 432, the data appear to be “partially” explained by lysosomal activity and iron metabolism.
The data presented in Tables S5 to S8 are very interesting.
On line 462, reword the sentence for clarity.
“As discussed above, an increase in soluble forms of iron was observed after MAG uptake ...”
On lines 470-474, revise the run-on sentence.
On line 494, refer to Figure 6A before referring to Figure 6B.
Very few cells display iron export activity through ferroportin (also known as SLC40A1). Use densitometry to help explain the western blot results.
DISCUSSION
This section needs a fair bit of editorial revision. The suggestions below are not all encompassing. Use these comments as a guideline for the needed revisions.
The first paragraph can be trimmed considerably and combined with the second paragraph. Avoid repetition.
On line 590, avoid speculating about results that were not quantitatively assessed.
Regarding text on line 593, it is not clear from the results presented here that MAG have much advantage over IONP. Isn’t it enough that MAG are comparable to IONP?
On line 599, I think you analyzed a monolayer of cells. What thin sections are you referring to?
On lines 610-612, revise the sentence as follows. There is no need to over speculate.
“... ICP-MS analysis provided a sensitive assessment of the trace release. Compared with IONPs, MAG exhibited faster iron ion release. This suggests that a small fraction of the MAG were degraded.
On line 614, do you mean “Fe3O4 encapsulated with ...”?
On lines 616-621, editorial revision is needed.
On line 633, reference 26 is the wrong citation.
The paragraph beginning on line 623 requires editorial revision.
On line 652, use densitometry to substantiate changes in protein expression.
On line 658, revise the wording.
“... is consistent with MAG degradation ...”
On line 664, revise the wording.
“... and the potential for regulation of signalling pathways, help to elucidate the response ...”
CONCLUSION
On line 676, if TEM is your only evidence of lysosomal compartmentalization, then refrain from firm conclusions about the nature of the vesicle until corroborating evidence can be presented.
Avoid dogmatic language. There are limitations of this study but the results are consistent with the potential for safely using MAG in biotechnology applications, particularly MRI where chemically synthesized SPIO have been approved for decades.
For example, on line 685, adjust the wording as follows.
“... MAG are biocompatible and should have excellent potential for cellular biotechnology and biomedicine applications.”
ACKNOWLEDGEMENTS
Fine
REFERENCES
Reference 16 is incomplete.
Check the author list in reference 26.
Reference 37 is a repeat of 36.
FIGURES & TABLES
Figure S1
What do the numbers in the figure represent?
Figure 1
On line 259 of the legend, the average IONP is 32 nm?
Table S1
Fine
Table S2
Fine
Table S3
In a footnote, indicate how amount of iron was measured.
Table S4
It would be helpful if a reference for the zeta potential measurement was provided in a footnote.
Figure 2
In this and all figure legends, use complete sentences to describe each part of the figure. Clearly indicate the experiment reported in each part.
What does the asterisk in part C represent?
Figure 3
In part A, add the scale bar to micrographs.
To help with orientation in parts B and C, label a few of the intracellular structures and indicate which part is MAG versus IONP.
Figure S2
Nice TEM
Figure 4
In part B, what do the dotted lines represent?
Figure S3
Describe the figure thoroughly in the figure legend.
Figure S4
Describe what the coloured dots refer to.
Figure 5
This reviewer appreciates the attempt at organizing the 158 up- or down-regulated proteins into categories.
Figure 6
In part C, is the carbonyl content any different in MAG-treated cells compared to the day 1 control?

Reviewer 2 Report
1. Last paragraph on page 7 and first paragraph on page 8 are repeated twice. Please make a correction.
2. Figure 3. Nanoparticle internalization by ECs. There are no captions in the TEM images of cells with MAG and IONPs in the figure 3(b) and (c). Also, there is no explanation available about figure 3(c) in the caption.
3. Why does the different shape and size of MAG and IONPs have different effect on cellular internalization time and process?
4. Does the regular arrangement of MAG inside the cell have any effect on degradation process? Could you please explain its significance?
5. How does this study will help in biomedical application? Could you give an example?
6. Could you explain the impact of immune cells like macrophages on the MAG nanoparticles?
Round 2
Reviewer 2 Report
No comments for the authors.